# Efficacy of Liraglutide in Obesity in Children and Adolescents: Systematic Review and Meta-Analysis of Randomized Controlled Trials

**DOI:** 10.3390/children10020208

**Published:** 2023-01-25

**Authors:** Alejandra Cornejo-Estrada, Carlos Nieto-Rodríguez, Darwin A. León-Figueroa, Emilly Moreno-Ramos, Cielo Cabanillas-Ramirez, Joshuan J. Barboza

**Affiliations:** 1School of Medicine, Universidad Peruana de Ciencias Aplicadas, Lima 15067, Peru; u201719365@upc.edu.pe (A.C.-E.); joaquin.nietor99@gmail.com (C.N.-R.); cabanillascielo0@gmail.com (C.C.-R.); 2Facultad de Medicina Humana, Universidad de San Martín de Porres, Chiclayo 15011, Peru; darwin_leon@usmp.pe; 3Unidad de Revisiones Sistemáticas y Meta-Análisis, Tau-Relaped Group, Trujillo 13007, Peru; emillynurse@gmail.com; 4Vicerrectorado de Investigacion, Universidad San Ignacio de Loyola, Lima 15046, Peru

**Keywords:** liraglutide, obesity, adolescents, children, systematic review

## Abstract

In the past few decades, childhood obesity has become a significant global health issue, impacting around 107.7 million children and adolescents globally. There is currently minimal usage of pharmacological therapies for childhood obesity in the pediatric population. This research assessed the efficacy of liraglutide in treating childhood and adolescent obesity. Until 20 October 2022, a systematic literature review was done utilizing PubMed, Scopus, Web of Science, and Embase databases. The search phrases “liraglutide”, “pediatric obesity”, “children”, and “adolescents” were utilized. Using the search method, a total of 185 articles were found. Three studies demonstrating liraglutide’s effectiveness in treating child and adolescent obesity were included. The selected research was done in the United States. As an intervention, liraglutide was administered to 296 participants at a maximal dosage of 3.0 mg. All examined trials were in phase 3. This comprehensive analysis revealed no clinically significant differences between liraglutide and body weight (kg; MD −2.62; 95%CI −6.35 to 1.12; *p* = 0.17) and body mass index (kg/m^2^; MD −0.80; 95%CI −2.33 to 0.73, *p* = 0.31). There was no evidence that liraglutide increased hypoglycemia episodes (RR 1.08; 95%CI 0.37 to 3.15; *p* = 0.79), or side consequences. However, it was shown that the medicine might help reduce BMI and weight combined with a healthy diet and regular exercise. A lifestyle change may have favorable consequences that will be assessed in the future for adjunctive therapy. PROSPERO database (CRD42022347472)

## 1. Introduction

The frequency of children and teenage obesity has skyrocketed in recent decades [1], impacting about 107.7 million children and adolescents worldwide [2], which has led to its classification as a public health issue [1,3,4,5]. Obesity is a complicated, recurring, progressive chronic condition [6], defined by an abnormal buildup of body fat [7,8]. This complex disorder of energy balance includes adipocytes, the gastrointestinal system, and the brain, by regulating food intake and energy expenditure [9]. 

Obesity is a significant risk factor contributing substantially to the global burden of major illnesses [10], where 1,1 billion adults and 10 percent of children are obese or overweight. This has resulted in a decrease in life expectancy [11,12].

Child obesity is connected with several health issues, including arterial hypertension (AHT), cardiovascular disease, type 2 diabetes, atherosclerosis, dyslipidemia, etc. [13,14,15]. In addition, childhood obesity can also harm the quality of life, resulting in negative psychological issues such as poor self-esteem, depression, and diminished school performance [16]. Currently, the treatment approaches for childhood obesity are multimodal [7], emphasizing promoting food and exercise-related lifestyle modifications [17,18].

The current application of medicine to manage obesity in the pediatric population is restricted [9]. In addition, the Food and Drug Administration (FDA) has authorized three medications for the treatment of childhood obesity: orlistat for patients less than 12 years, phentermine for those older than 16 years, and liraglutide for adolescents aged 12 to 17 years [19,20].

Liraglutide is a medication that is an analog of the intestinal incretin hormone GLP-1 generated by recombinant DNA technology; it is a very effective therapy for treating obesity [21,22,23]. As a response to the stimulus of food intake, the L cells of the intestine begin to secrete the intestinal incretin hormone GLP-1, which has the function of reducing blood glucose levels by inhibiting the secretion of glucagon by the alpha cells of the pancreas, while simultaneously generating an impulse to increase the release of insulin by the beta cells [24]. Moreover, liraglutide inhibits stomach motility and suppresses appetite via the central nervous system (CNS). The highest recommended dose of liraglutide for its indication against obesity is 3.0 mg per day; however, it is also marketed at a dose of 1.8 mg for diabetes [25]. Saxenda is the brand name for liraglutide [26].

Most existing pharmacological therapies for obesity have failed to demonstrate sufficient efficacy due to undesirable side effects, resulting in the discontinuation of several medicines for patient safety concerns [27]. Therefore, determining the effectiveness of liraglutide in treating child and teenage obesity was the purpose of the current investigation.

## 2. Materials and Methods

### 2.1. Eligibility Criteria

Included studies met the following: (1) Randomized controlled trials (RCT); (2) Pediatric patients (5–≤18 years) with a diagnosis of obesity as defined by the study authors, (3) Liraglutide (3.0 mg dose) as the intervention/experimental group and (4) Placebo or standard treatment control group. Studies meeting the following criteria will be excluded: Systematic reviews, meta-analysis, narrative reviews, editorials, abstracts, reports, and case series.

### 2.2. Information Sources and Search Strategy

A systematic search was carried out in PubMed, Scopus, Web of Science, and Embase. The search terms used were: (“Liraglutide”) AND (“Pediatric Obesity”) AND (“Children”) AND (“Adolescent”). Keywords and Mesh/Emtree thesauri were applied to establish search strategies for each database. The searches were completed on 20 October 2022, and two different investigators independently evaluated the search results.

### 2.3. Study Selection

Three researchers (A.C.E., C.N.R, and J.J.B.) generated a database based on electronic searches, administered using the proper management software (EndNote), and deleted duplicates. Then, using Rayyan Q.C.R.I. (https://rayyan.qcri.org/, last accessed on 22 October 2022), [28] two researchers (C.C.R., D.A.L.F.) performed the screening by separately assessing the titles and abstracts recovered by the search, selecting those that appeared to meet the eligibility criteria, and, if necessary, evaluating the full text. In the event of disagreement, the investigators will talk until they reach a consensus.

### 2.4. Extraction of Data

From the selected studies, the following data were extracted: author details, study type, country, number of participants per intervention arm, selection criteria, intervention and control description, and primary and secondary outcomes. In addition, a third researcher examined the list of publications and data extractions to confirm that there were no duplicate articles or material and addressed any inconsistencies about the inclusion of studies. 

### 2.5. Risk of Bias Analysis

Two writers (D.A.L.-F. and C.C.R.) separately evaluated the bias risk in randomized controlled trials using the Cochrane Risk of Bias 2.0 assessment; disagreements/differences were addressed by a discussion with a third author (J.J.B.).

### 2.6. Meta-Analysis

Inverse variance method and random effect meta-analyses were performed to compare the effects of Liraglutide and placebo on outcomes. The findings of meta-analyses were presented in the form of mean differences (M.D.) for continuous outcomes and relative risks (R.R.) for dichotomous outcomes. The Hartung-Knapp technique was utilized to update the 95% confidence intervals (C.I.s) for the effects, while the Paule-Mandel method was utilized to calculate the variance tau2 between trials. Heterogeneity was analyzed using the I^2^ index, with I^2^ 30% representing low heterogeneity, I^2^ 30–60% representing medium heterogeneity, and I^2^ > 60% representing high heterogeneity. For meta-analyses, the R meta-package was utilized. 

### 2.7. Assessment of the Reliability of the Evidence

The GRADE [29] guideline was utilized to evaluate the quality of the outcomes evidence. When evaluating the GRADE technique, the following factors were considered: risk of bias, imprecision, indirectness, inconsistency, and publication bias.

## 3. Results

### 3.1. Study Selection

After the search, a total of 185 articles were found. PRISMA-2020 flowchart (Figure 1) depicts the selection approach. After eliminating duplicates, the reviewers examined 123 articles. After filtering the titles and abstracts, 27 papers were selected for full-text evaluation, and three were considered acceptable for inclusion in this systematic review and meta-analysis [5,30,31].

### 3.2. Characteristics of the included studies

Of the 3 studies included in the present systematic review, all were conducted in the United States during the last decade with 2 years of difference between each one. The number of participants varied according to the criteria of each author respectively, in this systematic review there was a total number of 296 participants [5,30,31]. The type of intervention used in the 3 RCTs analyzed was through a dose of Liraglutide, with a maximum dose of 3.0 mg in all the studies. All the studies analyzed in the present investigation are randomized clinical trials that are in Phase 3. The three studies evaluated have conclusions that support each other due to the similar results found. It was possible to conclude in the three studies that treatment with Liraglutide in pediatric patients causes positive changes in their health as a consequence of its use, such as a reduction in body weight, BMI, and adverse episodes, providing results that support its tolerability and safety (Table 1) [5,30,31].

The risk of bias in the included studies was shown in Figure 2. The studies conducted by Mastrandrea (Mastrandrea et al., 2018) [5] and Danne (Danne et al, 2016) [31] were rated as poor quality with a risk of bias. On the other hand, the study by Kelly (Kelly et al., 2020) [30] was assumed to be of high-quality risk of bias.

### 3.3. Effect of Liraglutide on Primary Outcomes

Liraglutide does not decrease BMI compared to placebo (MD −0.8; 95%CI −2.33 to 0.73; *p* = 0.31; Figure 3). Similarly, Liraglutide does not decrease body weight compared to placebo (MD −2.62; 95%CI −6.35 to 1.12; *p* = 0.17; Figure 4).

### 3.4. Effect of Liraglutide on Secondary Outcomes

Liraglutide does not increase hypoglycemic episodes compared to placebo (RR 1.08; 95%CI 0.37 to 3.15; *p* = 0.79; Figure 5). Similarly, Liraglutide does not increase total adverse effects compared to placebo (RR 1.1; 95%CI 0.64 to 1.9; *p* = 0.52; Figure 6).

### 3.5. Certainty of Evidence

All outcomes in this systematic review have very low certainty (Table 2).

## 4. Discussion

Liraglutide was not associated with a reduction in pediatric patients’ body weight (kg), BMI (kg/m^2^), or blood pressure (mmHg) in this systematic study. Liraglutide was not seen to increase the frequency of hypoglycemia episodes or side effects. Although this is not the first systematic review and meta-analysis focusing on the effects of Liraglutide, it is one of the first to include the pediatric population (5–18 years) without concomitant morbidities, such as Diabetes Mellitus.

Due to its complicated pathophysiology, which involves socioeconomic, family, and individual variables [32], childhood obesity has become a serious worldwide health issue with challenging management [32,33,34]. Unfortunately, due to the intricate interplay of underlying determinants (nutrition, activity, and lifestyles), family and school-based programs treating childhood obesity have been adopted poorly [35,36].

The therapies utilized to treat childhood obesity are restricted and mainly consist of lifestyle modifications, medication, and bariatric surgery [37,38,39]. Therefore, lifestyle modification should be the first-line treatment for pediatric obesity, and in severe instances, further medication and bariatric surgery may be required [40].

Currently, there is little usage of pharmacological therapies for childhood obesity in the pediatric population, with inadequate evidence for each treatment choice [41]. These drugs are glucagon-like peptide-1 receptor-1 agonists (GLP-1RAs) designed to lower body weight in patients with type 2 diabetes mellitus (T2DM), and Liraglutide is the first GLP-1RA approved for the treatment of obesity in people with and without type 2 diabetes mellitus (T2DM) [40].

Chadda KR et al. reported that therapy with GLP-1 agonists lowered body weight more in children with obesity than in children with T2DM [42]. Therefore, increased GLP-1 secretion results in increased insulin secretion, boosting hepatic lipogenesis and promoting adipogenesis, which may explain the correlations of fasting GLP-1 with percent body fat, triglycerides, and alanine aminotransferase [43].

The research by T. Danne et al. offers evidence that the safety and tolerability profile of Liraglutide in adolescents with obesity was comparable to that in adults, with no unanticipated safety/tolerability concerns. Participants only received Liraglutide 3.0 mg for 1 to 2 weeks, making it impossible to establish the long-term effectiveness and safety of Liraglutide in this teenage group [31] due to the small sample size and short treatment duration.

The study by Mastrandrea LD et al. presents evidence of short-term therapy with Liraglutide in obese children, revealing a safety and tolerability profile comparable to studies in obese adults and adolescents, with no additional safety concerns. However, there is variation across treatment groups, and the experiment’s short duration means that the long-term safety could not be evaluated in this approximately 7-week trial [5].

The study by Kelly AS et al. provides evidence that lifestyle improvement is associated with weight control in teenagers. The administration of Liraglutide for 56 weeks showed no significant impact on cardiometabolic risk variables such as blood pressure, serum cholesterol, and triglycerides [30]. In addition, the participant’s energy consumption and physical activity were not examined; nevertheless, energy intake and physical activity are significant factors influencing body weight, and no data on these variables were reported at baseline and the conclusion of the research [30]. Therefore, metabolic profile abnormalities are prevalent in obese adolescents [44]. Lastly, Liraglutide side effects, particularly nausea and vomiting, may have caused weight loss [45].

Liraglutide showed a distinct effect on the adult population than on the pediatric group. Barboza JJ et al. reported that Liraglutide decreased body weight (mean difference (MD) −3.35 kg; 95% confidence interval (CI) −4.65 to −2.05; *p* < 0.0001) and BMI (mean difference (MD) −1.45 kg/m^2^; 95% confidence interval (CI) −1.98 to −0.91; *p* < 0.0001) in non-diabetic obese individuals [46]. Suhrs HE et al. showed that in overweight, non-diabetic women, 3 mg of Liraglutide per day for 12 weeks resulted in considerable weight reduction compared to baseline (absolute difference −6.03 kg; 95% confidence interval (CI) −5.22 to −6.84; *p* < 0.001) [47]. In addition, Zhang P et al. [48] showed that Liraglutide is an effective and safe medication for weight reduction in obese persons without Diabetes. 

The effective therapy of juvenile obesity requires a focus on evaluating the quality of life and avoiding obesity-related comorbidities [49] in addition to weight loss. The optimal therapy for juvenile obesity should be a medicine that is highly tailored and customized, taking into account age, coexisting disorders, drug tolerance, and local and economical medical situations [50,51]. 

### Limitations

This systematic review has many limitations. First, few RCT studies evaluating the effectiveness of Liraglutide in obese children were discovered. Second, the studies did not demonstrate beneficial impacts. Many had small sample sizes, short-term follow-ups, or needed a control group, making it difficult to evaluate the findings across research. Thirdly, neither trial was primarily designed to assess the time-limited BMI decrease at maximal dosage. In addition, none of these clinical trials contained a component regarding lifestyle.

## 5. Conclusions

In this systematic review presented we did not find a clinically significant difference concerning body weight (kg), and BMI (kg/m^2^) with the use of liraglutide. Nor was liraglutide found to increase hypoglycemic episodes or adverse effects. However, it was observed that the drug can help lower BMI and weight accompanied by adequate diet and exercise. In adjunctive therapy, it may have positive repercussions that will be evaluated in the future with a lifestyle change.

## Figures and Tables

**Figure 1 children-10-00208-f001:**
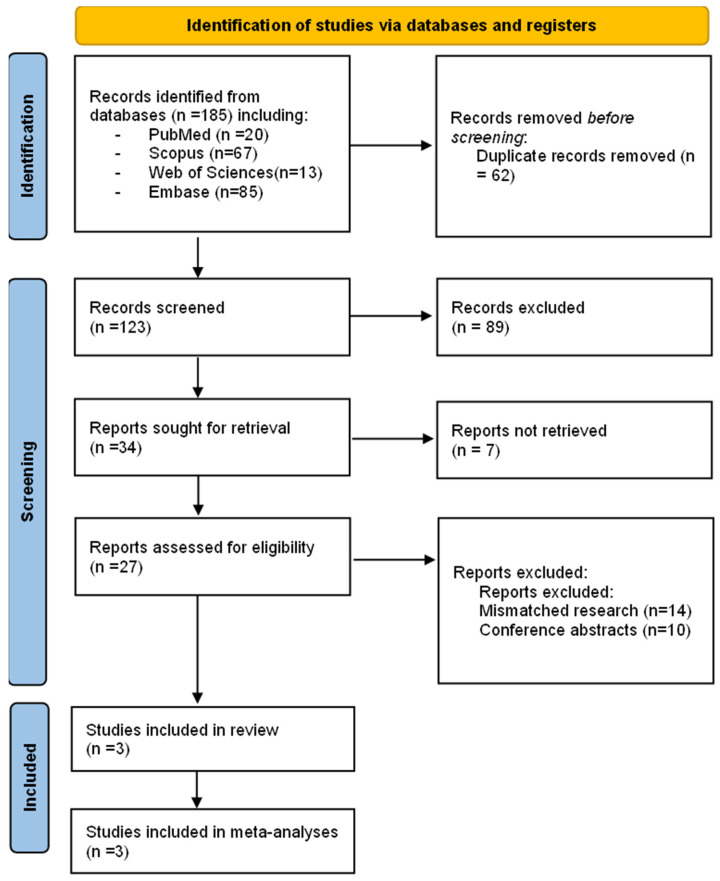
PRISMA flow chart of the studies selection process.

**Figure 2 children-10-00208-f002:**
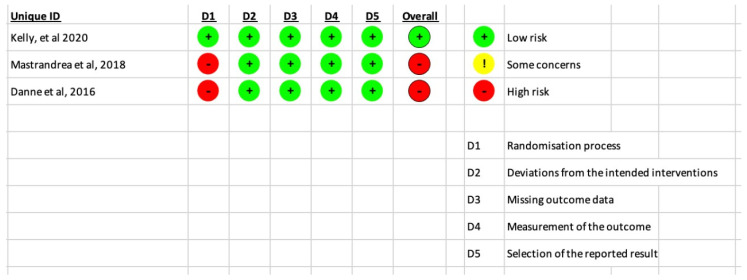
Results of the risk of bias analysis for the included RCTs [5,30,31].

**Figure 3 children-10-00208-f003:**
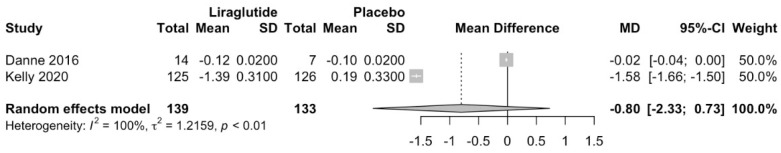
Effect of Liraglutide on BMI [30,31].

**Figure 4 children-10-00208-f004:**
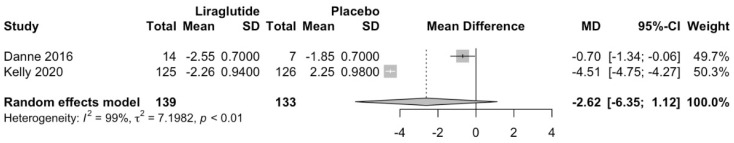
Effect of Liraglutide on Body Weight [30,31].

**Figure 5 children-10-00208-f005:**
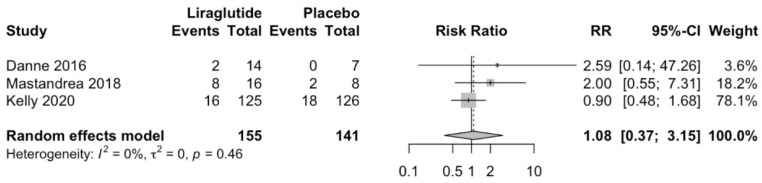
Effect of Liraglutide in hypoglycemic episodes [5,30,31].

**Figure 6 children-10-00208-f006:**
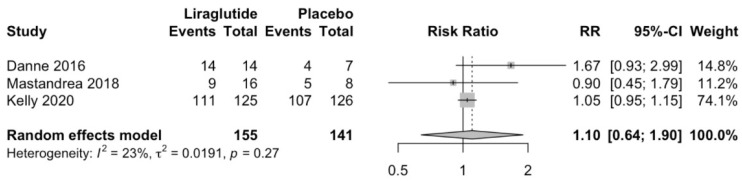
Effect of liraglutide on total adverse effects [5,30,31].

**Table 1 children-10-00208-t001:** The main characteristics of the included studies.

Author	Year of Publication	Country	Trial Registration	Participants (n)	Trial Phase	Type of Intervention	Type of Control	Primary Outcome	Secondary Outcome
Danne T, et al. [31]	2016	United States	NCT01789086	21	Phase 3	Liraglutide 0.6 mg with weekly dose increase up to a maximum of 3.0 mg during the last week.	Placebo	BMI score (MD,SD): Liraglutide: −0.12 (0.02) Placebo: −0.10 (0.02) Body weight (BW,SD): Liraglutide: −2.55 (0.70) Placebo: −1.85 (0.70)	Adverse events (n, %): Liraglutide 14/14 (100%). Placebo 4/7 (57.1) Nausea (n, %): Liraglutide 7 (50%) Placebo 0 (0%) Gastrointestinal adverse event: Liraglutide 11 (78.6) Placebo 0 (0%) Hypoglycemic episodes: Liraglutide 2 (14.3%) Placebo 0 (0%)
Mastrandrea LD, et al. [5]	2018	United States	NCT02696148	24	Phase 3	Liraglutide was increased from 0.3 mg to 1.2 mg in weekly increments of 0.3 mg and then followed with weekly increments of 0.6 mg up to a maximum dose of 3.0 mg.	Placebo	Total adverse events (n, %): Liraglutide 9/16 (56.3%). Placebo 5/8 (62.5)	Nausea (n, %): Liraglutide 2 (12.5%). Placebo 0 (0%) Gastrointestinal adverse event: Liraglutide 6 (37.5%) Placebo 1 (12.5) Hypoglycemic episodes: Liraglutide 8 (50%) Placebo 2 (25%)
Kelly AS, et al. [30]	2020	United States	NCT02918279	251	Phase 3	Liraglutide 3.0 mg	Placebo	BMI score (MD,SD): Liraglutide: −1.39 (0.31) Placebo: 0.19 (0.33) Body weight (BW,SD): Liraglutide: −2.26 (0.94) Placebo: 2.25 (0.98)	Adverse events (n,%): Liraglutide 111/125 (88.8%). Placebo 107/126 (84.9%) Nausea (n,%): Liraglutide 53 (42.4%). Placebo 18 (14.3%) Gastrointestinal adverse event (n,%): Liraglutide 81 (64.8%) Placebo 46 (36.5%) Hypoglycemic episodes (n,%): Liraglutide 26 (20.8%) Placebo 18 (14.28%)

**Table 2 children-10-00208-t002:** Summary of findings, GRADE certainty of evidence.

Liraglutide Compared to Placebo for Obesity in Children and Adolescent					
Outcomes	No. of Participants (Studies) Follow-Up	Certainty of the Evidence (GRADE)	Relative Effect (95% CI)	Anticipated Absolute Effects	
				Risk with Placebo	Risk Difference with Liraglutide
Body mass index (BMI) assessed with: kg/cm^2^	272 (2 RCTs)	⨁◯◯◯ Very low ^a,b,c^	-	The mean body mass index was 0.09 kg/m^2^	MD 0.8 kg/m^2^ lower (2.33 lower to 0.73 higher)
Body Weight assessed with: kg	272 (2 RCTs)	⨁◯◯◯ Very low ^a,b,c^	-	The mean body Weight was 0.4 kg	MD 2.62 kg lower (6.35 lower to 1.12 higher)
Hypoglycemic episodes assessed with: Frequency	296 (3 RCTs)	⨁◯◯◯ Very low ^a,d^	RR 1.08 (0.37 to 3.15)	142 per 1000	11 more per 1000 (89 fewer to 305 more)
Total adverse effects assessed with: Frequency	296 (3 RCTs)	⨁◯◯◯ Very low ^a,b,e^	RR 1.10 (0.64 to 1.90)	823 per 1000	82 more per 1000 (296 fewer to 740 more)

CI: confidence interval; MD: mean difference; RR: risk ratio. ^a^ Decreases two levels due to high risk of bias; ^b^ Decreases two levels for imprecision; ^c^ Decreases two levels for inconsistency; ^d^ Decreases one level for imprecision; ^e^ Decreases one level for inconsistency.

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
