# Peer review of "Efficacy of Liraglutide in Obesity in Children and Adolescents: Systematic Review and Meta-Analysis of Randomized Controlled Trials"

_children, 2023, doi:10.3390/children10020208_

Round 1
Reviewer 1 Report
GLP 1 receptor agonists (RA) in the treatment of pediatric obesity is very timely and clinically relevant thus this is an important study.
Line 59- stated there are only 2 medications that are FDA approved for pediatric obesity- there are 3 medications including liraglutide which was approved for adolescents aged 12-17 in 2020
Is liraglutide the only GLP 1RA of interest since there are other GLP1RA that are being used as well pending insurance approval.
Please clarify classification of obesity as stated below, additionally classification below is primarily used in adults although can be countered to be used in children, the pediatric classification in children of BMI percentiles were not considered or BMI zscore (although not used as much anymore). Additionally, would’ve been interesting to look at effect on severe obesity.
In 46 class 0 they have a BMI of 20-25 kg/m2 and are not considered obese; In class I they have 47 a BMI of 25-30 kg/m2 and have a low risk of obesity; Class II patients have a BMI of 30-35 48 kg/m2 and have a moderate risk; Class III patients have a BMI of 35-40 kg/m2 and have a 49 high risk associated with their obesity for disease; Class IV individuals have a BMI greater 50 than 40 kg/m2 and have a very high risk of disease [11].
It might be helpful to cite some adult data in the introduction to give a comparison to what was found in children
Author Response
- Reviewer comment: GLP 1 receptor agonists (RA) in the treatment of pediatric obesity is very timely and clinically relevant thus this is an important study.
Our response: “Thank you very much for your comments.”
- Reviewer comment: Line 59- stated there are only 2 medications that are FDA approved for pediatric obesity- there are 3 medications including liraglutide which was approved for adolescents aged 12-17 in 2020.
Our response: “FDA-approved drugs were corrected. Thank you very much for the clarification and recommendation.”
- Reviewer comment: Is liraglutide the only GLP 1RA of interest since there are other GLP1RA that are being used as well pending insurance approval.
Our response: “Information was better organized according to published studies. They were described in the introduction.”
- Reviewer comment: Please clarify classification of obesity as stated below, additionally classification below is primarily used in adults although can be countered to be used in children, the pediatric classification in children of BMI percentiles were not considered or BMI zscore (although not used as much anymore). Additionally, would’ve been interesting to look at effect on severe obesity.
Our response: “Information was sought and the classification of obesity was drafted based on published articles.”
- Reviewer comment: In 46 class 0 they have a BMI of 20-25 kg/m2 and are not considered obese; In class I they have 47 a BMI of 25-30 kg/m2 and have a low risk of obesity; Class II patients have a BMI of 30-35 48 kg/m2 and have a moderate risk; Class III patients have a BMI of 35-40 kg/m2 and have a 49 high risk associated with their obesity for disease; Class IV individuals have a BMI greater 50 than 40 kg/m2 and have a very high risk of disease [11].
Our response: “Thank you very much for the clarification and recommendation regarding the BMI classification.”
- Reviewer comment: It might be helpful to cite some adult data in the introduction to give a comparison to what was found in children.
Our response: “Thank you very much for the recommendation. Relevant information was added in the discussion section regarding some adult data to give a comparison with what was found in children”

Reviewer 2 Report
Although this manuscript is a timely topic evaluating the use of pharmacotherapy, this paper is premature as there are only three RCTs to evaluate and only one is a year-long RCT designed to measure weight reduction. I do not think based on the available literature that the authors can drawn meaningful conclusions and that the results of the Kelly et al. RCT is diluted by short term (5 to 13 week) studies not designed to measure BMI reduction. The Cochrane handbook suggests a minimum of at least 6-10 studies so I don't think a meaningful or sound conclusion can be drawn from this current analysis.
Additionally, information in the introduction is not up to date or accurate; areas for corrections are below:
1. Line 60: there are 2 additional FDA-approved medications for pediatric obesity medicine besides phentermine and orlistat that the authors list: liraglutide 3.0mg (approved in 2020) and phentermine/topiramate (approved in 2022).
2. Line 66. Liraglutide 3.0mg also slows gastric motility and has CNS-mediated appetite suppression that should be listed as mechanisms.
3. Line 71. The maximum recommended dose of liraglutide is 3.0mg/day for its anti-obesity indication, but it is also marketed at a 1.8mg dose for anti-diabetes use.
4. Line 46. BMI as an absolute value is not how pediatric obesity is categorized as healthy BMI varies by age and gender. Rather the convention is to classify by percentage of the 95th percentile. I.e. Class II obesity is between 1.2 x and 1.4 x the 95th percentile and class III obesity is > 1.4x the 95th percentile for BMI.
5. Line 111. In pediatrics BP range is based on height percentile not weight.
The primary aim of the Mastrandrea et al. and Danne et al papers were treatment-emergent AEs. Neither study was primarily designed to measure BMI reduction with limited time on the maximum dose. This should be listed by the authors as a limitation and ultimately should disqualify them from the meta-analysis. Furthermore there was no lifestyle component to either of these trials so the statement that "most of these studies found on the pharmacological treatment of pediatric obesity [lira 3.0mg only in actuality] are accompanied by lifestyle co-interventions (line 250) is untrue.
The study by Kelly et al. did contain a lifestyle intervention for both the treatment and placebo groups. Lifestyle alone (placebo group) had minimal weight change so the estimated treatment difference of −4.64%, favoring liraglutide does evaluate the efficacy of the drug vs lifestyle alone (line 252).
Author Response
Although this manuscript is a timely topic evaluating the use of pharmacotherapy, this paper is premature as there are only three RCTs to evaluate and only one is a year long RCT designed to measure weight reduction. I do not think based on the available literature that the authors can draw meaningful conclusions and that the results of the Kelly et al. RCT is diluted by short term (5 to 13 wweek) studiesnot designed to measure BMI reduction. The Cochrane handbook suggests a minimum of at least 6-10 studies so I don't think a meaningful or sound conclusion can be drawn from this current analysis.
Additionally, information in the introduction is not up to date or accurate; areas for corrections are below:
- Reviewer comment: Line 60: there are 2 additional FDA-approved medications for pediatric obesity medicine besides phentermine and orlistat that the authors list: liraglutide 3.0mg (approved in 2020) and phentermine/topiramate (approved in 2022).
Our response: “Thank you for your comments. Corrected the FDA-approved medications”
- Reviewer comment: Line 66. Liraglutide 3.0mg also slows gastric motility and has CNS-mediated appetite suppression that should be listed as mechanisms.
Our response: “The suggestion provided was added. Allowing to improve the focus of the information presented”
- Reviewer comment: Line 71. The maximum recommended dose of liraglutide is 3.0mg/day for its anti-obesity indication, but it is also marketed at a 1.8mg dose for anti-diabetes use.
Our response: “The information provided was added. Thank you very much.”
- Reviewer comment: Line 46. BMI as an absolute value is not how pediatric obesity is categorized as healthy BMI varies by age and gender. Rather the convention is to classify by percentage of the 95th percentile. I.e. Class II obesity is between 1.2 x and 1.4 x the 95th percentile and class III obesity is > 1.4x the 95th percentile for BMI.
Our response: “Thank you very much for your comments and observations. The BMI classification for the pediatric population has been corrected.”
- Reviewer comment: Line 111. In pediatrics BP range is based on height percentile not weight.
Our response: “Corrected based on the BMI classification for the pediatric population provided by the CDC.”
- Reviewer comment: The primary aim of the Mastrandrea et al. and Danne et al papers were treatment-emergent AEs. Neither study was primarily designed to measure BMI reduction with limited time on the maximum dose. This should be listed by the authors as a limitation and ultimately should disqualify them from the meta-analysis. Furthermore, there was no lifestyle component to either of these trials so the statement that "most of these studies found on the pharmacological treatment of pediatric obesity [lira 3.0mg only in actuality] are accompanied by lifestyle co-interventions (line 250) is untrue.
The study by Kelly et al. did contain a lifestyle intervention for both the treatment and placebo groups. Lifestyle alone (placebo group) had minimal weight change so the estimated treatment difference of −4.64%, favoring liraglutide does evaluate the efficacy of the drug vs lifestyle alone (line 252).
Our response: “The approach was corrected according to their recommendations. In addition, a section on limitations was added to emphasize what was suggested.”

Reviewer 3 Report
I am only suggest add a couple references of this topic
Obes Rev
. 2021 Jun;22(6):e13177. doi: 10.1111/obr.13177. Epub 2020 Dec 22.
GLP-1 agonists for obesity and type 2 diabetes in children: Systematic review and meta-analysis
Editorial
J Clin Endocrinol Metab
. 2021 Aug 18;106(9):e3778-e3780. doi: 10.1210/clinem/dgab301.
Elevation of Fasting GLP-1 Levels in Child and Adolescent Obesity: Friend or Foe?
Author Response
- Reviewer comment: I am only suggest add a couple references of this topic Obes Rev. 2021 Jun;22(6):e13177. doi: 10.1111/obr.13177. Epub 2020 Dec 22.
GLP-1 agonists for obesity and type 2 diabetes in children: Systematic review and meta-analysis. Editorial. J Clin Endocrinol Metab. 2021 Aug 18;106(9):e3778-e3780. doi: 10.1210/clinem/dgab301. Elevation of Fasting GLP-1 Levels in Child and Adolescent Obesity: Friend or Foe?.
Our response: “Thank you very much for your recommendations. They were added in the discussion section of the shared articles.”

Reviewer 4 Report
The manuscript "Efficacy of Liraglutide in obesity in children and adolescents: systematic review and meta-analysis of randomized controlled trials" by Cornejo-Estrada et al analyses the efficacy and adverse outcomes of Liraglutide in the pediatric population. The article describes new and relevant insights into a novel anti-obesity treatment in very young patients without diabetes or other comorbidities.
There are a few comments to mention:
1. Eligibility criteria for study inclusion: meta-analysis were also excluded?
2. Study selection: there are some verbs that are in future tense instead of past tense.
3. All study outcomes could be mentioned in the study objectives at the end of the Introduction chapter
4. The discussion chapter could be enriched with data regarding the effect of Liraglutide use in the adult population with obesity.
Author Response
Reviewer comment: The manuscript "Efficacy of Liraglutide in obesity in children and adolescents: systematic review and meta-analysis of randomized controlled trials" by Cornejo-Estrada et al analyses the efficacy and adverse outcomes of Liraglutide in the pediatric population. The article describes new and relevant insights into a novel anti-obesity treatment in very young patients without diabetes or other comorbidities.
There are a few comments to mention:
- Reviewer comment: Eligibility criteria for study inclusion: meta-analysis were also excluded?
Our response: “For our systematic review, we excluded meta-analysis studies. We added in the exclusion terms.”
- Reviewer comment: 2. Study selection: there are some verbs that are in future tense instead of past tense.
Our response: “Thank you very much for your comments. Some grammar mistakes have been corrected.”
- Reviewer comment: All study outcomes could be mentioned in the study objectives at the end of the Introduction chapter.
Our response: “The most salient results are found in the first section of the discussion. Thank you very much for your recommendation.”
- Reviewer comment: The discussion chapter could be enriched with data regarding the effect of Liraglutide use in the adult population with obesity.
Our response: “Excellent recommendation. Added high-impact studies evaluating the effects of liraglutide use in the adult population with obesity.”

Round 2
Reviewer 2 Report
Thank you for addressing my concerns. As previously stated, this paper is premature as there are only three RCTs to evaluate and only one is a year long RCT designed to measure weight reduction. I do not think based on the available literature that the authors can draw meaningful conclusions and that the results of the Kelly et al. The combined results of this metanalysis is diluted by short term (5 to 13 week) studies not designed to measure BMI reduction. The Cochrane handbook suggests a minimum of at least 6-10 studies so I don't think a meaningful or sound conclusion can be drawn from this current analysis and therefore this manuscript should not be described as a meta-analysis.
Author Response
Dear Reviewer,
There is no consensus, because the performance of a meta-analysis does not depend on the number of studies to be analyzed, but on the assumptions that the studies must meet in order to be meta-analyzed.
Criteria for performing a meta-analysis.
✅Outcomes are comparable and can be weighted (forms of measurement + measures of effect + units of measurement must be equal).
✅ Interventions and controls must be measured in the same form/measure, or at least must be sufficiently similar.
✅Data to be analyzed must be available. For binary data the events in the experimental group and control group should be considered. For continuous data, the mean (aka. final mean) and standard deviation should be considered.
✅If there are continuous data, these should be able to be calculated in case the authors of the included studies do not provide complete data.
✅Have at least TWO studies included in the systematic review that meet all of the above criteria.
These are not criteria that exclude the performance of a meta-analysis.
❌Having fewer than 3 included studies.
❌Have heterogeneity greater than 60% in I2.
Reference:
Ryan R; Cochrane Consumers and Communication Review Group. 'Cochrane Consumers and Communication Group: meta-analysis.' http://cccrg.cochrane.org, December 2016.